# Characterization of CNC Nanoparticles Prepared via Ultrasonic-Assisted Spray Drying and Their Application in Composite Films

**DOI:** 10.3390/nano13222928

**Published:** 2023-11-10

**Authors:** Sungjun Hwang, Yousoo Han, Douglas J. Gardner

**Affiliations:** 1Advanced Structures and Composites Center, University of Maine, 35 Flagstaff Road, Orono, ME 04469-5793, USA; douglasg@maine.edu; 2School of Forest Resources, University of Maine, 5755 Nutting Hall, Orono, ME 04469-5755, USA

**Keywords:** cellulose nanocrystals, nano spray dryer, polyvinyl alcohol

## Abstract

The ultrasonic-assisted spray dryer, also known as a nano spray dryer and predominantly used on a lab scale in the pharmaceutical and food industries, enables the production of nanometer-sized particles. In this study, the nano spray dryer was applied to cellulosic materials, such as cellulose nanofibrils (CNFs) and cellulose nanocrystals (CNCs). CNC suspensions were successfully dried, while the CNF suspensions could not be dried, attributable to their longer fibril lengths. The nano spray drying process was performed under different drying conditions, including nebulizer hole sizes, solid concentrations, and gas flow rates. It was confirmed that the individual particle size of nano spray-dried CNCs (nano SDCNCs) decreased as the nebulizer hole sizes and solid contents of the suspensions decreased. The production rate of the nano spray dryer increased with higher solid contents and lower gas flow rates. The resulting nano SDCNCs were added to a polyvinyl alcohol (PVA) matrix as a reinforcing material to evaluate their reinforcement behavior in a plastic matrix using solvent casting. After incorporating the 20 wt.% nano SDCNCs into the PVA matrix, the tensile strength and tensile modulus elasticity of the neat PVA nanocomposite film increased by 22% and 32%, respectively, while preserving the transparency of the films.

## 1. Introduction

Cellulose is a linear, high-molecular-weight natural polymer consisting of repeating β-(1→4)-D-glucopyranose units. It is predominantly found in the secondary cell walls of both woody and nonwoody cellulosic plants [1,2]. Typically, the width of natural fibrils is measured in micrometers, and they can be reduced to nanometer-scale fibrils using mechanical, chemical, or biological methods [3]. Using a chemical treatment, cellulose nanocrystals (CNCs) can be obtained through strong acid hydrolysis [4]. CNCs are immersed and stirred in a sulfuric acid solution. After the centrifugation process, only the supernatant is collected. Subsequently, dialysis is employed to remove excess water [5]. CNCs exhibit a rod-like shape with diameters ranging from 20 to 60 nm and lengths ranging from 250 to 480 nm. CNCs are known for their beneficial properties, including renewability, biodegradability, low density, oxygen gas barrier capabilities, and dimensional stability [6,7].

There are three common types of spray drying methods: ultrasonic spray dryer, two-fluid nozzle system, and rotary disk atomizer [8]. The two-fluid nozzle uses high compressed gas pressure, and the rotary disk atomizer employs centrifugal energy to atomize suspensions [9,10]. During the drying process, individual fibers agglomerate, forming particles as the liquid solution evaporates. For a more detailed understanding of the evaporation process, the droplets begin to evaporate rapidly in the drying chamber, achieving their maximum evaporation rate. This phase is referred to as the constant-rate drying period. During this phase, water from the interior of the droplet migrates to its surface, ensuring the surface remains saturated and allowing evaporation to continue at a consistent rate. When the rate of evaporation from the surface becomes faster than the rate at which water can migrate from inside the droplet to the surface, the constant-rate drying period ends. At the critical moisture content following the constant-rate drying period, the droplet cannot maintain a saturated surface attributable to the reduced water migration rate in the droplet. As a result, the droplet surface becomes unsaturated. This transition marks the beginning of the falling-rate drying period. In this phase, a solid layer, referred to as a crust, forms around the droplet. This crust impedes the continuity of the diffusion, causing a decrease in the evaporation rate and an increase in the temperature of the particle. After the evaporation process, dried particles then collide with the inside wall of the cyclone, losing kinetic energy, and subsequently fall into the collector [11,12,13,14,15,16,17]. Based on the particle collection system, the conventional types of spray dryers cannot produce nanoscale individual particles. This is because particles less than 2 μm, which are lighter, can exit the dryer chamber along with the exhaust gas, while heavier particles larger than 2 μm are deposited in the collector [16].

An ultrasonic-assisted spray dryer referred to as the nano spray dryer (B-90) is used in various industries, including pharmaceuticals, materials, and food sciences [17,18,19]. The nano spray dryer employs a piezoelectric-driven vibrating membrane in the spray head, which provides an ultrasonic frequency of 80–140 kHz [20]. This drying technique produces aerosol droplets ranging from 300 nm to 5 μm, emanating from the nebulizer hole in the spray head [21]. Furthermore, instead of a cyclone, the nano spray dryer uses an electrostatic collector. This leads to effective particle adhesion to the sidewalls of the drying chamber, subsequently increasing production yield. The inlet temperature in the nano spray dryer increases quickly up to 120 °C through a heater providing a laminar gas flow. The utilization of laminar gas flow enhances the drying efficiency compared to the turbulent airflow used in the two-fluid nozzle and rotary disk atomizer [22]. The turbulent airflow leads to uncontrolled spray formation and particle loss [23]. In addition, the nano spray dryer operates efficiently with a very small amount (e.g., 2 mL) of liquid as feedstock [19]. Furthermore, the outlet temperature of the nano spray dryer is notably lower compared to the other two spray dryers [21]. This makes it particularly suitable for drying raw materials in heat-sensitive and high-value products [24].

As shown in Figure 1, the drying gas is heated up by a heater module in the nano spray dryer that produces laminar flow. The suspended solids are continuously fed to the spray head via a peristaltic pump. The thin spray mesh containing an array of laser-drilled holes in the center of the spray head vibrates upwards and downwards at ultrasonic frequencies (80–140 kHz). Fine droplets are ejected from the nebulizer hole in the spray head. Then, the water in the droplets is evaporated by hot air, followed by the remaining cellulose fibrils becoming agglomerated, resulting in particle formation. The particles fall toward the particle collector. The particle collector consists of two electrodes: a stack of star-shaped anodes positioned at the center and a steel circular cathode on the sidewall of the drying chamber. When high voltage (15 kV) is applied to the electrodes, the electrical potential difference increases, triggering a corona discharge on the sharp points of the stack of star-shaped anodes. After the water in the droplets evaporates, the dried particles become negatively charged from electrostatic attachment resulting from the corona discharge. Subsequently, the negatively charged particles move toward the positively charged collecting electrode because of the Coulomb force [19,20,21,22,23,24,25,26,27].

Polyvinyl alcohol (PVA) is a water-soluble, hydrophilic polymer that has been used in various industries, including food packaging, composite films, filtration membranes, ion-battery separators, etc. The prevalent use of PVA can be attributed to its advantageous properties, including easy processability, nontoxicity, transparency, gas barrier capability, and biodegradability in aerobic environments. Furthermore, the hydroxyl groups on PVA can form hydrogen bonds with hydroxyl groups on CNCs in a water medium, increasing the mechanical properties of CNC-reinforced PVA composites [28,29,30,31,32,33,34,35,36]. The mechanical properties of PVA can be further enhanced through cellulose modification or cross-linking treatments with the utilization of chemicals [37]. However, the use of chemicals is harmful to the environment.

Previous studies used conventional spray dryers with two-fluid nozzles and a rotary disk atomizer for spray drying cellulosic materials, including cellulose nanofibrils (CNFs) and cellulose nanocrystals (CNCs); however, there is limited research on producing nanometer-sized particles using CNFs and CNCs with a nano spray dryer. While similar work was conducted by Sanders [38], he produced nanosized CNC particles using an electrospray drying (ESD) technique but not a nano spray dryer. This study aims to explore the feasibility of using CNFs and CNCs in a nano spray dryer to produce individual nanometer-sized particles. Attempts to use CNF suspensions in the nano spray dryer were unsuccessful. In contrast, the successful production of nano spray-dried CNC powders was achieved using the nano spray dryer. The morphological properties, particle size distribution, and production rates of the nano spray-dried CNCs (nano SDCNCs) were determined under various drying conditions in the nano spray dryer, including nebulizer hole sizes, solid contents, and gas flow rates. After the nano SDCNC powders were added to the PVA matrix as a reinforcing material via solvent casting, the visual inspection, ultraviolet-visible (UV-Vis) spectra, and tensile properties of the resulting PVA composite films were analyzed. Spray-dried CNC powder, dried using a two-fluid nozzle, was used as the control. It is important to note that the characteristics of the nanocomposite film were assessed to determine the viability of incorporating CNC nanoparticles into composite film applications.

## 2. Materials and Methods

### 2.1. Spray Drying Conditions

CNF and CNC suspensions were provided by the Process Development Center (PDC) at the University of Maine (Orono, ME, USA) and the U.S. Forest Products Laboratory (Madison, WI, USA). The solid contents of the CNF and CNC suspensions were 3.3 wt.% and 10.3 wt.%, respectively. CNF and CNC suspensions were dried using a Buchi B-90 laboratory nano spray dryer (New Castle, DE, USA). Table 1 shows the experimental parameters for the nano spray drying process. The nebulizer hole sizes of 4.0, 5.5, and 7.0 μm were used, and the solid contents of CNC suspensions were adjusted to 1.0, 3.0, and 5.0 wt.%. The gas flow rates of 100 and 120 L/min were adjusted. The nano spray dryer has a maximum temperature limit of 120 °C. For this study, the inlet temperature was set to this maximum to optimize water evaporation from the droplets and reduce residual moisture in the dried particles. The residual moisture in dried powders leads to strong sticking on the walls of the drying chamber [17,22]. In general, controlling the peristaltic pump rate in the nano spray dryer affects droplet size and feed rate. The water feed rate can be defined by the peristaltic pump setting; 50% corresponds to about 900 mL/h in the mini spray dryer. In contrast, the water feed rate in the nano spray dryer can be varied by the different nebulizer hole sizes; 10–20 mL/h (4.0 µm spray mesh), 25–50 mL/h (5.5 µm), and 80–150 mL/h (7.0 µm). If the feed is pumped at a higher rate, it increases the pressure on the spray mesh inside the spray cap, leading to an increase in initial droplets and the feed rate [22]. In this study, when the pump rate exceeded 30%, high feed rates led to overflow issues, while the CNC suspension did not reach the spray head at pump rates under 30%. As a result, a 30% pump rate was fixed in this study. Spray intensity reflects the duration the suspension remains in the spray mesh while vibrating, and it can be adjusted from 10% to 100%. Lower intensities lead to reduced throughput, prolonging the overall process, while increasing the intensity produces more droplets per time unit. However, it only marginally affects the droplet size [22]. The spray intensity was set at 80% in this study. When the setting exceeds 80%, an exclamation mark ‘!’ appears on the display, indicating a potential quicker reduction in the spray mesh span life. After adjusting various process conditions for the nano spray dryer, nano SDCNCs produced under optimal conditions were chosen as the reinforcing material in a PVA matrix. Optimal conditions in this study were characterized by a relatively high production rate and the capability to produce nanometer-sized particles. The production rate was determined by weighing the powders after four hours of the spray drying process and then reporting it as the rate per hour. Spray-dried CNC powder, produced using a Buchi B-290 laboratory mini spray dryer (New Castle, DE, USA), served as a control to highlight the distinct properties of nano SDCNCs. The conditions applied in the mini spray dryer were an inlet temperature of 175 °C, a gas flow rate of 90 L/min, an aspirator rate of 100%, and a pump rate of 8 mL/min (Table 2). The resulting mini spray-dried CNCs are referred to as mini SDCNCs in this study.

### 2.2. Solvent Casting

A powder form of fully hydrolyzed polyvinyl alcohol (PVA) with an average molecular weight of 145,000 was purchased from Sigma-Aldrich (St. Louis, MO, USA). Figure 2 depicts the solvent-casting procedure used to produce PVA nanocomposite films. The PVA powders were dissolved in distilled water to achieve 1.5 wt.% solid content suspensions and were then stirred vigorously at 90 °C for 40 min. This PVA solution was subsequently mixed with 20 wt.% nano SDCNC powders. The same content of mini SDCNCs was used for comparison with the nano SDCNC-reinforced PVA nanocomposite films (Table 3). After stirring the mixtures for an additional 40 min, they were poured into glass dishes with a radius of 4.75 cm. Once the distilled water in the mixture had evaporated in a laminar air flow hood over 48 h, the films were gently detached from the glass substrates. The target basis weight for each film was set at 60 g/m^2^. Five nanocomposite films were produced under each set of process conditions.

### 2.3. Characteristics of Spray-Dried CNC Powders

The scanning electron microscope (SEM) micrographs of CNC particles were obtained using a Zeiss NVision 40 (Seiko Instruments Inc., Chiba, Japan), with an accelerating voltage set of 3 kV. The specimens were coated with Au/Pd to a thickness of 8 nm. The individual particle size was determined using Image-J software (Image-J 1.53e, Montgomery, MD, USA) based on the SEM images. The particle size distribution (PSD) and the surface area [D3,2] values were measured using the Mastersizer 2000 (Malvern Instruments, Worcestershire, UK). A total of 1 g of powder was scooped and then placed on the tray in the Scirocco 2000 attachment (Malvern, UK), and the powder was analyzed with a particle refractive index of 1.53. The surface area [D3,2] value was also considered as the average particle size of the sample. This is also referred to as Sauter mean diameter, defined by Equation (1) [39,40]:(1)$$\;d_{3.2}=\frac{1}{\sum_{i}\frac{p_{i}}{d_{i}}}=\frac{6}{\mathrm{specific}\;\mathrm{area}}$$

*d_i_*: Mean diameter of class *i*;

*P_i_*: Relative volume probability of class *i*.

The aspect ratio and HS Circularity of CNC particles were measured using a Morphologi-G3-ID morphologically directed optical microscope system (Malvern Instruments, Worcestershire, UK). The two equations below represent HS Circularity Equation (2) and aspect ratio Equation (3) in the Morphologi-G3 [41].
(2)$$\mathrm{HS}\;\mathrm{Circularity}=\frac{4\pi \mathrm{Area}}{{\mathrm{Perimeter}}^{2}}$$
(3)$$\mathrm{Aspect}\;\mathrm{Ratio}=\frac{\mathrm{Width}}{\mathrm{Length}}$$

The SEM micrographs of PVA nanocomposite films were obtained via the Hitachi Tabletop Microscope SEM TM 3000 (Hitachi High-Technologies Corporation, Tokyo, Japan). The accelerating voltage was 15 kV and various magnifications were adjusted automatically.

### 2.4. Characteristics of PVA Nanocomposite Films

Five PVA nanocomposite films for each condition were conditioned at 23 °C and 50% RH for 24 h before any tests, according to TAPPI T 402 [42]. The basis weight (the weight per unit area) of the PVA nanocomposite films was measured according to TAPPI T 220 [43]. The tensile strength and tensile modulus of elasticity of the nanocomposite films (Appendix A) were determined using a universal testing machine (Instron 5564, Norwood, MA, USA) with displacement control loading at a loading speed of 1 in/min and a load cell of 500 N. The samples were 15 mm wide, and the span length between two clamping jaws was adjusted to be 68 mm.

The transparency of the PVA nanocomposite films was observed through a visual inspection and a UV-vis Spectrophotometer (Perkin Elmer Instruments, Lambda 365, Waltham, MA, USA). The reference used in spectra was air with a wavelength range of 400 to 900 nm.

## 3. Results and Discussion

### 3.1. Feasibility of Spray Drying

The attempt to dry cellulose nanofibrils (CNFs) using a nano spray dryer was unsuccessful. Despite the use of low solids content (~0.1 wt. %) and a large nebulizer hole size, no droplets were ejected from the nebulizer hole. The CNFs used for spray drying contained an average fiber length of 304 μm from a 90% fines level of bleached Kraft pulp refined by a thermal disk refiner [44]. The fines level represents the percentage of fibers under 200 μm in the total fiber count, measured with the MorFi Fiber Analyzer (TechPap, Grenoble, France). The long fiber length of the CNFs led to entanglement in the nebulizer hole, quickly blocking the spray head (Figure 3). It is worth noting that the spray head can be blocked if the particle size exceeds 1/10 of the nebulizer hole diameter [24]. Figure 4a shows an image of a 12 wt.% aqueous slurry of CNC suspensions, displaying a white-colored aqueous gel. Figure 4b shows a transmission electron microscopy (TEM) image of CNCs, rod-like or whisker-shaped, sized 5–20 nm in width and 150–200 nm in length [34,45]. As shown in Figure 5, CNC suspensions were successfully dried using a nano spray dryer. The tiny droplets of the CNC suspension were ejected from the membrane in the nebulizer hole.

### 3.2. Effects of Spray Drying Conditions on Particle Shapes and Sizes of Nano SDCNCs

The drying conditions including solids content of CNC suspensions, nebulizer hole size, and gas flow rate in the nano spray dryer were varied to assess their impact on particle shapes, particle size distribution, and production rates after the spray drying process. Two methods were used to measure the average size of nano SDCNC particles. The average particle size of scooped powders was measured using the surface area [D3,2] value from the Mastersizer 2000. Meanwhile, Image-J software (Image-J 1.53e, Montgomery, MD, USA) determined the average size of individual particles. Notably, the average particle size observed from Image-J was consistently smaller than the [D3,2] values for all samples. This discrepancy is likely because the Mastersizer 2000 detects agglomerates or clusters of CNC particles, while Image-J measures only individual CNC particles. As a result, the [D3,2] values indicate the presence of agglomerates or lumps, leading to larger average particle sizes.

Figure 6 shows the SEM images of CNC powders dried using a nano spray dryer, varying by the nebulizer hole sizes. All other drying conditions remained consistent. After spray drying, fine CNC powders with a circular shape were produced for all nebulizer hole sizes. Image-J analysis showed that the average individual particle sizes were 811 nm, 1.4 µm, and 2 µm for particles ejected from small, medium, and large nebulizer holes, respectively. The [D3,2] values indicated average particle sizes of 1.1 µm, 6.0 µm, and 5.6 µm for those ejected from the small, medium, and large nebulizer holes, respectively (Figure 7). It can be noted that the [D3,2] values of the resulting particles produced from a large nebulizer hole showed no difference from the medium nebulizer hole. However, the particle size distribution graph resulting from the large nebulizer hole, exhibited a broader range, especially toward larger size portions, compared to that from the medium nebulizer hole. Overall, both Image-J and Mastersizer 2000 indicated that the average particle size and particle size distribution of nano SDCNCs decreased and became narrower, respectively, as the nebulizer hole diameter was reduced. This is likely attributable to the fact that a smaller mesh aperture produces finer droplets of CNC solutions (Figure 8) [47]. Notably, nanometer-sized SDCNC particles were produced only from the small nebulizer hole.

Figure 9 shows the SEM images of CNC powders dried using a nano spray dryer with variations in the solids content of the CNC suspensions, while other drying conditions remained consistent. In this study, the CNC suspension concentration was limited to 5.0 wt.%. Exceeding 5.0 wt.% led to the blockage of the small nebulizer hole by the agglomerated CNC fibrils, similar to the CNF agglomeration discussed earlier. In the Image-J analysis, individual particle sizes of nano SDCNCs were observed as 530 nm, 811 nm, and 896 nm for concentrations of 1.0 wt.%, 3.0 wt.%, and 5.0 wt.%, respectively. The [D3,2] values indicated sizes of 0.9 µm, 1.1 µm, and 1.8 µm for the same concentrations (Figure 10). Both the Image-J and [D3,2] values indicated that the average particle sizes decreased as the solid contents of the CNC suspensions decreased. The lower solids content resulted in a smaller number of CNC fibers per unit volume of the droplets during the spray drying process, leading to smaller agglomerates of CNC fibers (Figure 11) [48,49]. The high viscosity and surface tension of the solution with a higher solids concentration reduced the feed rate (amount of fluid sprayed per unit of time) at the same peristaltic pump rates, slowing the discharge of droplets from the vibrating mesh holes, resulting in the increase in the droplet size. The larger droplets encapsulate a greater amount of CNCs, increasing particle size (Figure 12) [22,49,50]. It is noteworthy that a small CNC particle (538 nm) was produced using a 1.0 wt.% CNC suspension and SDCNC particles were measured less than 1 µm, despite being determined from the [D3,2] value. The average individual SDCNC particles were twice the size of the dried CNC particles, which were 208 nm in length and 78 nm in width, via electrospray drying (ESD) conducted by Sanders et al. 2023 [38].

As shown in Figure 13 and Figure 14, the drying gas flow rate between 100 L/min and 120 L/min in the nano spray dryer did not significantly affect the particle shapes and particle size distribution. This is consistent with the findings of previous research [8,21,48,49].

### 3.3. Effects of Spray Drying Conditions on Production Rates

Figure 15 shows the production rates of nano SDCNCs after drying in the nano spray dryer according to the different nebulizer hole sizes, solid contents, and gas flow rates. Each measurement was obtained after a consistent drying period of 1 h. For the small (#1 in Table 1), medium (#5), and large (#6) nebulizer holes, the production rates were 200 mg, 300 mg, and 140 mg, respectively. In addition, the production rates for 1 wt.% (#2), 3 wt.% (#1), and 5 wt.% (#3) of CNC suspensions were 120 mg, 200 mg, and 110 mg, respectively. The production rate at the higher gas flow rate (120 L/min) was reduced by half compared to that at the lower gas flow rate (100 L/min). In general, the feed rate can be increased with a larger mesh size and lower solids concentration [22,27]. Large mesh holes allow for a greater volume of suspensions, and a lower solids concentration accelerates the discharge of droplets from the vibrating mesh holes [22,27,51]. Increasing the feed rate in the dryer leads to a decreased evaporation rate attributable to the introduction of more water. This results in a higher moisture content in the final product. The increased moisture causes the powders to become stickier, leading to greater deposits on the drying chamber walls, reducing production rates [52]. A higher production yield was reported by previous research with a reduction in the feed rate [53]. A high air flow rate enhances moisture removal during drying. However, if the air is humid, a higher flow rate might be less effective and can introduce more moisture into the resulting powders [8,22]. In this study, the results show a tendency that is somewhat like previous research, but they did not completely match. It is worthwhile to note that variations in production rates might occur attributable to particle depositions around the spray cap, deposited sticky powders on the chamber walls, and losses during the manual collection of the powder [22,27]. A rubber scraper was used to detach ultra-fine powders from the chamber wall, allowing them to fall onto the paper surface, and the production rates were determined.

### 3.4. Nano Spray Dryer vs. Mini Spray Dryer

Figure 16a,c shows the SEM images of nano SDCNCs produced using a small nebulizer hole with a 3.0 wt.% solids content of CNC suspension. Individual nanometer-sized CNC particles were observed with a relatively high production rate at the #1 drying condition (Table 1). The resulting nano SDCNCs were compared to mini SDCNCs dried by a mini spray dryer (Figure 16b,d), and the solids content of CNCs was the same as 3.0 wt.%. Because the droplet size is directly affected by the velocity level of the atomizing gas [10,54], the gas flow rate was set to 540 L/h based on the previous research conducted by Peng et al**.** [55], as the smallest individual CNC particles were produced at this gas flow rate. The average individual particle sizes of the nano SDCNCs and mini SDCNCs observed by an Image-J were 811 nm and 1.8 µm, respectively, indicating that the nano SDCNCs were 55% smaller than that of the mini SDCNCs. In addition, the [D3,2] value of the nano SDCNCs was 1.1 µm, which is 59% smaller compared to the 2.7 µm of the mini SDCNCs (Figure 17). Figure 18 shows the aspect ratio and HS Circularity of nano and mini SDCNCs. The resulting value of the aspect ratio in Morphologi-G3 is presented reciprocally to the commonly accepted interpretation of the aspect ratio. The closer their value to one in HS Circularity indicates, the closer the shape of the circle. The aspect ratio values of the nano SDCNCs and mini SDCNCs, ranging from 0.8 to 1, were represented as percentages of 40% and 34%, respectively. Similarly, for the HS Circularity, with values ranging from 0.8 to 1, the nano SDCNCs and mini SDCNCs were 57% and 24%, respectively. Overall, the nano SDCNCs contained much smaller individual particles with a lower aspect ratio and more spherical shape compared to the mini SDCNCs. In the mini spray dryer, the kinetic energy of the high-velocity gas flow serves as an atomization source, shattering the bulk liquid into ligaments that subsequently break into droplets. According to previous research, the mini spray dryer provides water droplets with an average size of approximately 14 µm, while the average droplet of water was approximately 4.8 µm when ejected from a small nebulizer hole size in the nano spray dryer [17]. Smaller droplets in a nano spray dryer contain fewer CNC fibers, attributable to their reduced capacity to hold a volume of CNC fibers. This reduces the number of fibers to be aggregated during drying, leading to a smaller average particle size with higher circularity and a lower aspect ratio. In addition, the use of the laminar gas flow and electrostatic particle collector in the nano spray dryer ensures uniform drying, without disturbance, maintaining the circular-shaped particles. In contrast, the turbulent gas flow and cyclone collector system in the mini spray dryer might lead to irregularly shaped particles after collisions between the particles themselves and the collector surface during spray drying. It can be worthwhile to report that the nano SDCNCs contained more circular-shaped particles compared to the CNC particles dried via electrospray drying, conducted by Sanders et al., 2023 [38]. Table 4 shows the spray-dried CNFs and CNCs produced using various atomization techniques as conducted in previous research studies.

### 3.5. Optical, Physical, and Mechanical Properties of PVA Nanocomposite Films

Figure 19 shows pictures of neat PVA and SDCNC-filled PVA nanocomposite films after solvent casting. Visual observation of the pictures revealed no noticeable difference in transparency between the neat PVA and the SDCNC-filled PVA nanocomposite films. The letters behind the films were clearly visible. In addition, the agglomeration of CNC particles was not visually detected. The values of transmittance in the visible light range (400–800 nm) were consistent between the neat PVA and mini and nano SDCNC-filled PVA nanocomposite films (Figure 20). PVA is known as a transparent polymer, with a transmittance exceeding 90% at a wavelength of 700 nm. The transparency of PVA is not significantly affected by the addition of mini and nano SDCNC particles. This is likely attributable to the interaction between the hydrophilic crystalline cellulose and the PVA matrix, resulting in high compatibility between the two materials [58,59]. Furthermore, the large-scale agglomeration of CNC particles might be prevented, attributable to their fine particle size.

Figure 21a shows the tensile strength and tensile modulus of elasticity of SDCNC-reinforced PVA nanocomposite films, and Figure 21b represents the corresponding stress–strain curve. The tensile strength and modulus of elasticity of neat PVA slightly decreased with the addition of mini SDCNCs. In contrast, incorporating nano SDCNCs into the PVA matrix results in increases of 26% and 32% in tensile strength and modulus of elasticity, respectively. The improved tensile properties of PVA nanocomposites can be attributed to the generation of hydrogen bonding between hydroxyl groups on the surface of SDCNCs and PVA chains, forming a relatively stable structure [58,59]. The distribution degree of the filler within the plastic matrix is a crucial factor influencing the tensile properties of the composites. Poor distribution of fillers can detrimentally affect the mechanical properties of the plastic matrix, as filler agglomeration hinders stress transfer between the filler and the matrix [60,61]. Moreover, higher filler loading can lead to the agglomeration of the fillers, negatively affecting the mechanical properties of the plastic composites [62,63]. The larger mini SDCNCs, consisting of individual particles exceeding 2 µm, and with a filler content of 20 wt.%, tend to form larger clumps, thereby decreasing the tensile properties of PVA nanocomposites. Conversely, nano SDCNCs, with individual particles smaller than 1 µm, possess a greater surface area that leads to an enhanced potential for bonding with the hydroxyl groups on PVA chains. In addition, the smaller particles of nano SDCNCs lead to a reduction in both the size and amount of filler agglomeration within the PVA matrix. It can be noted that the excellent distribution of nano SDCNC powders in the PVA matrix enhances the tensile properties of PVA nanocomposite films, even with the addition of 20 wt.% filler contents (Figure 22).

The advantageous effects of spray-dried CNF powders into thermoplastic matrices are well addressed in our previous research [56,64,65]. The dispersion and distribution issues of fillers into plastic matrices were enhanced by adding spray-dried CNF powders. The main difference between previous research and this study was the choice of solvent casting rather than melt-compounding. It is believed that dispersion can be enhanced by spray drying cellulose fibrils, demonstrating the reinforcing effect of SDCNC powders into the PVA matrix.

## 4. Conclusions

The production of nanoparticles of CNFs and CNCs was attempted by using an ultrasonic-assisted spray dryer. It failed to produce dry particles using CNF suspensions because of the blockage of the long fiber length, while nanosized CNC particles were successfully produced. Nanopowders of CNC were produced using the nebulizer with the smallest hole size, and the particles, measuring 530 nm, were produced through a 1.0 wt.% solids content of CNC suspension. The size of the nano spray-dried CNC powders decreased with smaller nebulizer hole sizes and lower solid contents of CNC suspension. The production rate was higher at 3.0 wt.% solids content of CNC suspension than that of 1.0 wt.% and was higher at 100 L/min compared to 120 L/min. After adding 20 wt.% nano spray-dried CNC particles into the PVA matrix, the tensile strength and tensile modulus of elasticity increased by up to 26% and 32%, respectively. This increase was attributable to the excellent dispersion and distribution of the very fine fillers into the plastic matrix. In addition, the transparency of the neat PVA was not compromised, and severe agglomerations of particles were not detected. In the future, further characterization of physical and mechanical properties will be conducted on the nano SDCNC-filled PVA composite films.

## Figures and Tables

**Figure 1 nanomaterials-13-02928-f001:**
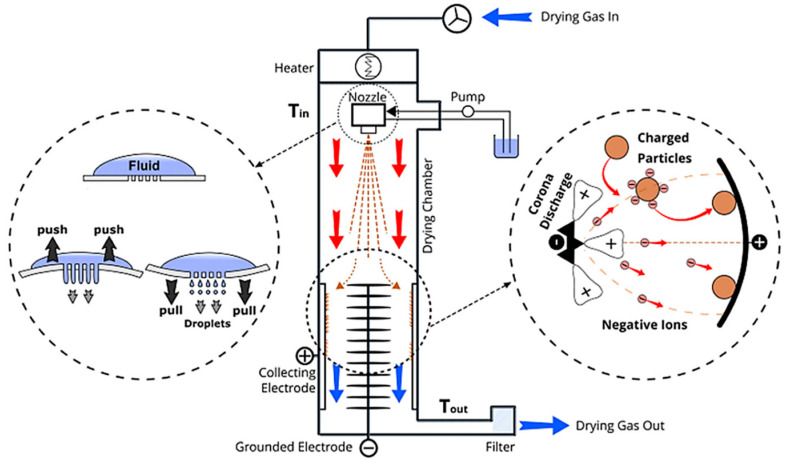
Schematic diagram of nano spray dryer (B-90) [25].

**Figure 2 nanomaterials-13-02928-f002:**
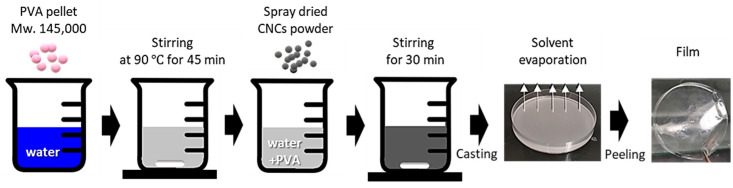
Procedure for PVA film solvent casting.

**Figure 3 nanomaterials-13-02928-f003:**
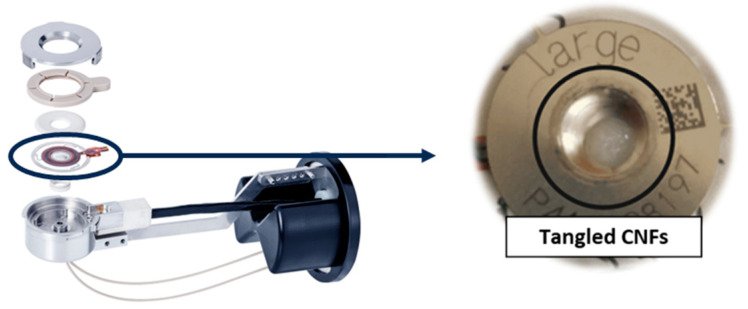
Picture of tangled cellulose nanofibrils in the large nebulizer hole in the spray head (modified based on [46]).

**Figure 4 nanomaterials-13-02928-f004:**
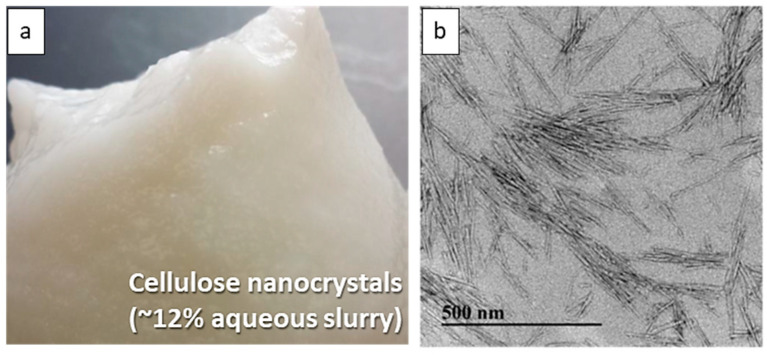
Micrographs of CNC slurry (**a**) and TEM image of CNCs (**b**) [34,45].

**Figure 5 nanomaterials-13-02928-f005:**
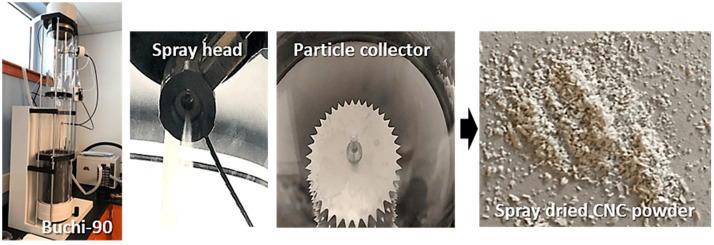
Production of CNC nanoparticles dried through a nano spray dryer.

**Figure 6 nanomaterials-13-02928-f006:**
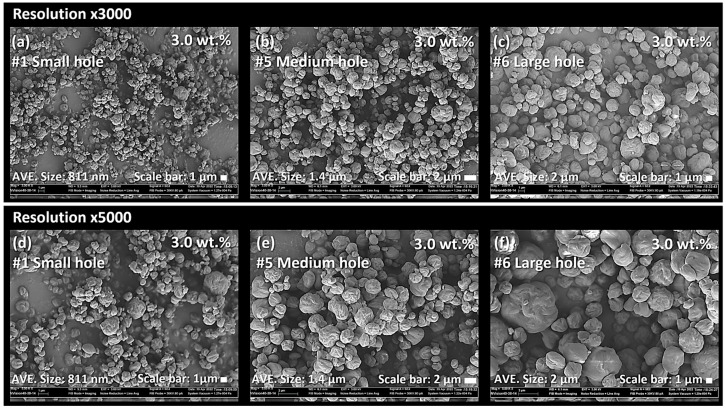
SEM micrographs of CNC nanoparticles dried using a nano spray dryer based on the different nebulizer hole sizes in the spray head: (**a**,**d**) small nebulizer hole, (**b**,**e**) medium nebulizer hole, and (**c**,**f**) large nebulizer hole.

**Figure 7 nanomaterials-13-02928-f007:**
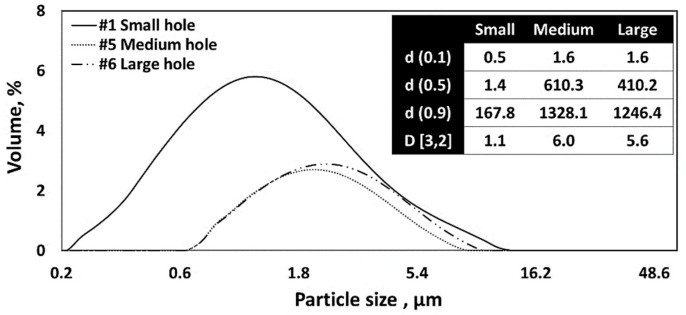
Particle size distribution of CNC nanoparticles dried using a nano spray dryer based on the different nebulizer hole sizes in the spray head.

**Figure 8 nanomaterials-13-02928-f008:**
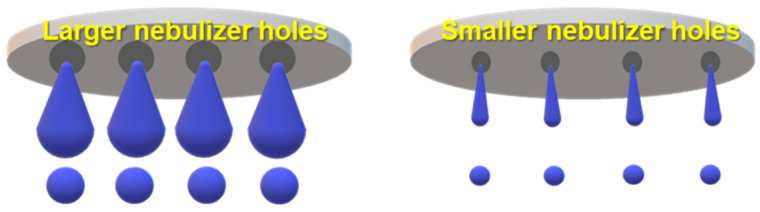
Schematic diagram of droplet size variation with different nebulizer hole sizes.

**Figure 9 nanomaterials-13-02928-f009:**
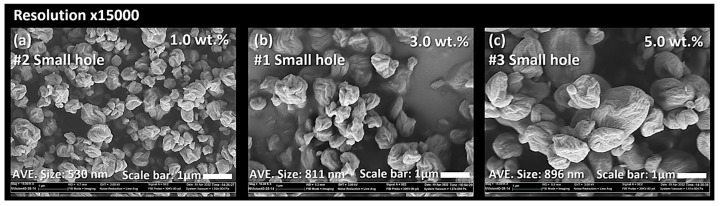
SEM micrographs of CNC nanoparticles dried using a nano spray dryer based on the different solid contents of 1.0 wt.% (**a**), 3.0 wt.% (**b**), and 5.0 wt.% (**c**) CNC suspensions.

**Figure 10 nanomaterials-13-02928-f010:**
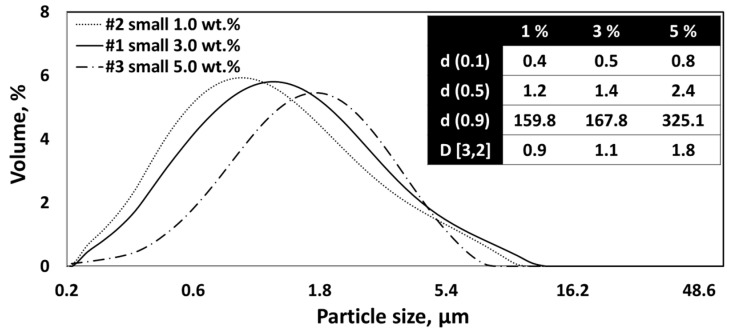
PSD of CNC nanoparticles dried using nano spray dryer based on the different solid contents of CNC suspensions.

**Figure 11 nanomaterials-13-02928-f011:**
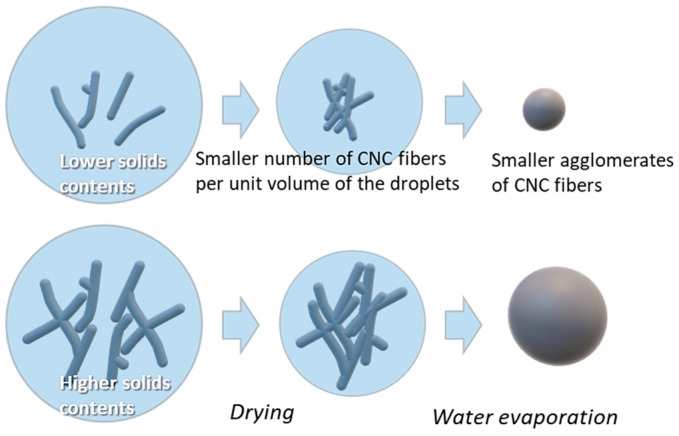
Schematic diagram of particle size variation with different solid contents in the CNC suspensions.

**Figure 12 nanomaterials-13-02928-f012:**
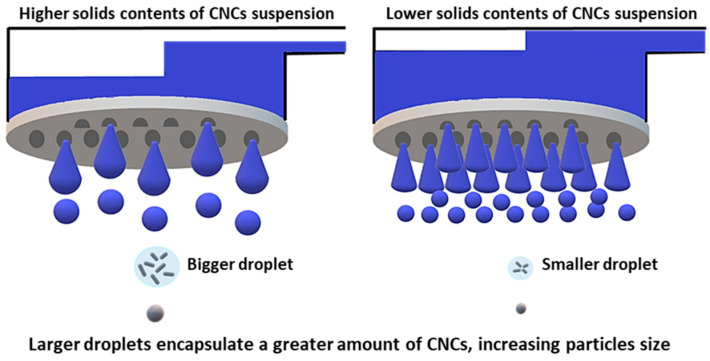
Schematic diagram of droplet size variation with different solid contents in the CNC suspensions.

**Figure 13 nanomaterials-13-02928-f013:**
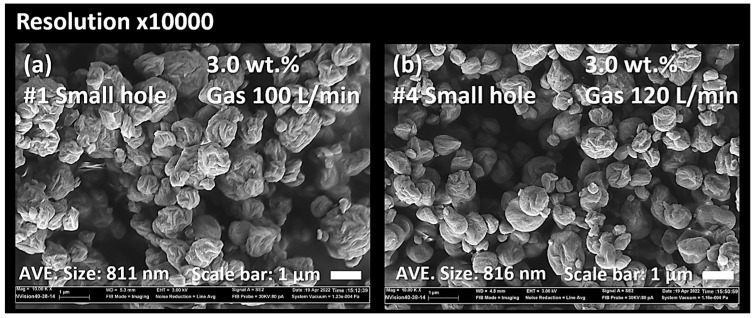
SEM micrograph of CNC nanoparticles dried using the nano spray dryer based on the different gas flow rates of 100 L/min (**a**) and 120 L/min (**b**).

**Figure 14 nanomaterials-13-02928-f014:**
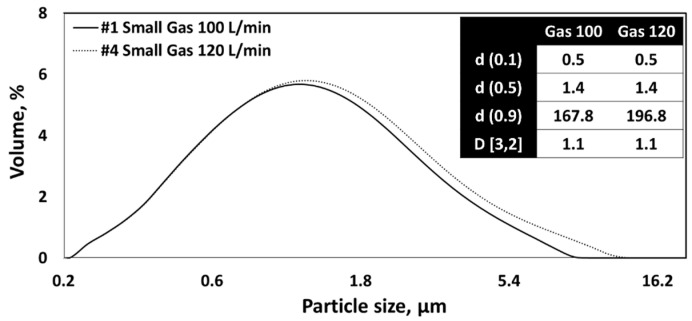
PSD of CNC nanoparticles dried using a nano spray dryer based on the different gas flow rates.

**Figure 15 nanomaterials-13-02928-f015:**
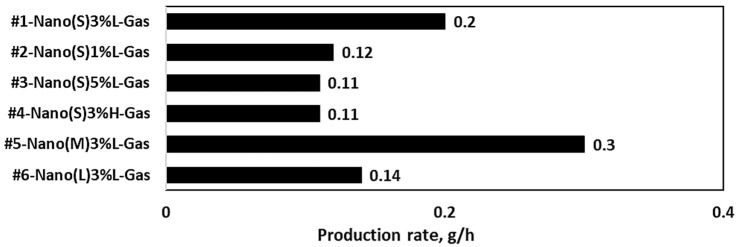
Production rates of a nano spray dryer.

**Figure 16 nanomaterials-13-02928-f016:**
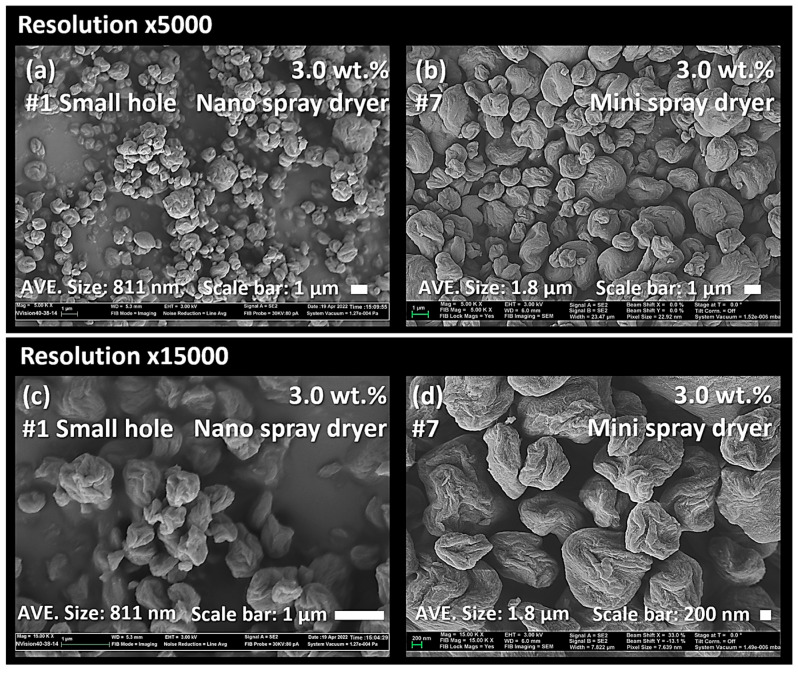
SEM micrograph of CNC nanoparticles dried using a nano spray dryer (**a**,**c**) and mini spray dryer (**b**,**d**).

**Figure 17 nanomaterials-13-02928-f017:**
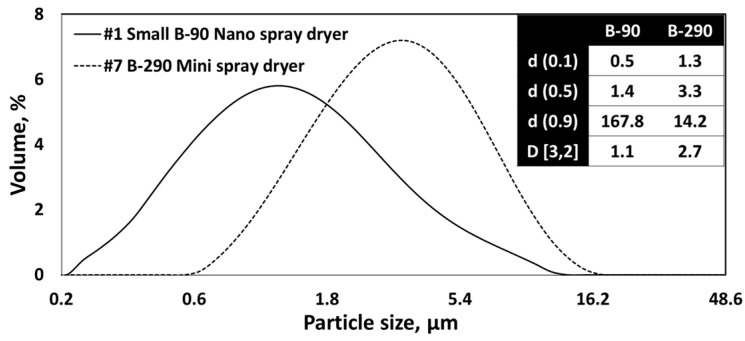
PSD of CNC nanoparticles dried using a nano spray dryer and mini spray dryer.

**Figure 18 nanomaterials-13-02928-f018:**
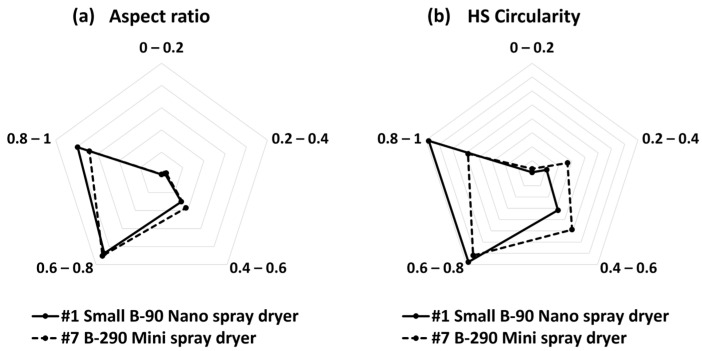
Aspect ratio (**a**) and HS Circularity (**b**) of SDCNCs dried though a nano spray dryer and mini spray dryer.

**Figure 19 nanomaterials-13-02928-f019:**
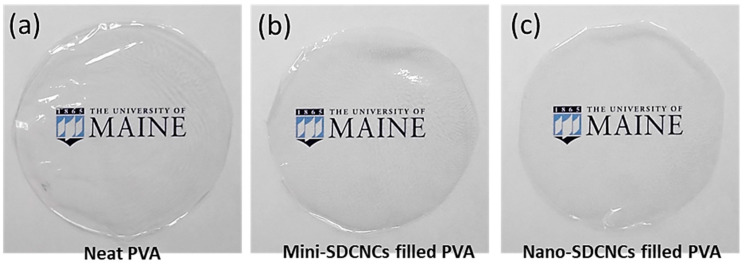
Pictures of neat PVA (**a**), mini SDCNC-filled PVA (**b**), and nano SDCNC-filled PVA (**c**).

**Figure 20 nanomaterials-13-02928-f020:**
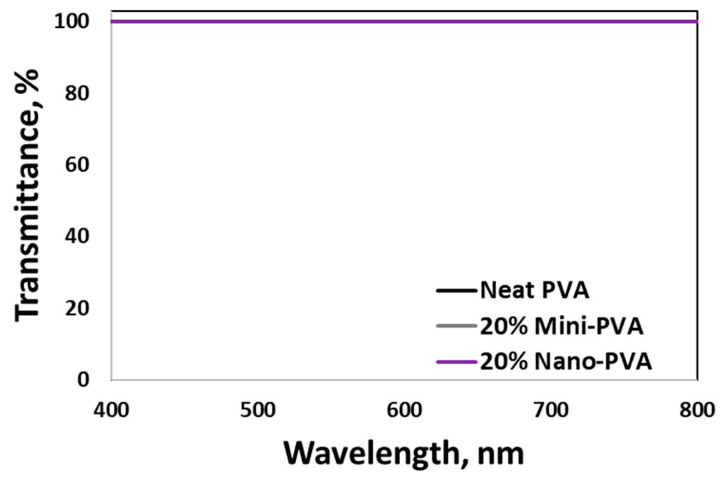
UV-vis transmittance of PVA nanocomposite films.

**Figure 21 nanomaterials-13-02928-f021:**
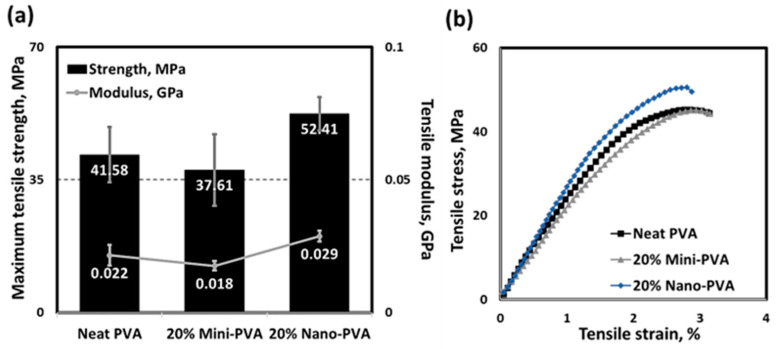
Tensile strength and tensile modulus of elasticity of PVA nanocomposite films (**a**) and strength–strain curve (**b**).

**Figure 22 nanomaterials-13-02928-f022:**
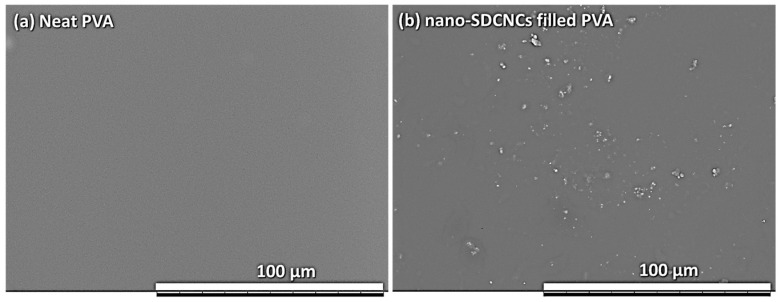
SEM micrographs of neat PVA film (**a**) and 20 wt.% nano SDCNC-filled PVA film (**b**).

**Table 1 nanomaterials-13-02928-t001:** Drying conditions of the nano spray dryer (B-90).

No.	Nebulizer Hole Size, (µm)	Solid Contents, (wt.%)	Inlet Temperature, (°C)	Gas Flow Rate, (L/min)	Spray Intensity, (%)
1	small (4 µm)	3	120	100	80
2	small (4 µm)	1	120	100	80
3	small (4 µm)	5	120	100	80
4	small (4 µm)	3	120	120	80
5	medium (5.5 µm)	3	120	100	80
6	large (7 µm)	3	120	100	80

**Table 2 nanomaterials-13-02928-t002:** Drying conditions of the mini spray dryer (B-290).

No.	Solid Contents, (wt.%)	Inlet Temperature, (°C)	Gas Flow Rate, (L/h)	Aspirator Rate, (%)	Pump Rate, (mL/min)
7	3	175	540	100	8

**Table 3 nanomaterials-13-02928-t003:** Solvent-casting formulation for PVA films.

Composite Film	Drying Technique	PVA Contents, wt.%	Filler Contents, wt.%
Neat PVA	-	100	-
Nano SDCNCs filled PVA	Nano spray dryer	80	20
Mini SDCNCs filled PVA	Mini spray dryer	80	20

**Table 4 nanomaterials-13-02928-t004:** Average particle sizes of spray-dried CNFs and CNCs depending on different feedstocks and drying techniques.

Feedstock	Drying Techniques	Average Particle Sizes	References
CNCs	Electrospray drying	208 nm in length, 78 nm in width	Sanders et al., 2023 [38]
CNCs	Ultrasonic-assisted spray dryer	530–896 nm in width	This study
CNCs	Two-fluid nozzle	CE diameter: 2.7–4.59 µm	This study Peng et al., 2012 [55]
CNFs	Two-fluid nozzle	CE diameter: 4.09–4.35 µm	Peng et al., 2012 [55]
CNFs	Pilot-scale rotary disk atomizer	CE diameter: 9.58–19 µm	Hwang et al., 2023 [56] Wang et al., 2018 [57]

CE: circle equivalent.

## Data Availability

All of the material is owned by the authors and/or no permissions are required.

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
