# Peer review of "Characterization of CNC Nanoparticles Prepared via Ultrasonic-Assisted Spray Drying and Their Application in Composite Films"

_nanomaterials, 2023, doi:10.3390/nano13222928_

Round 1

Reviewer 1 Report

Comments and Suggestions for Authors

The article proposed by Hwang and others, describes the use of a specific instruments for the production of ultrasonic spary-dried cellulose nanoparticles. Sometimes this work appeared rather a technical report than a scientic paper since it is related to some parameters that are typical of the instrument that has been used. For example in more that one occasione the authors setted the parameter to 80 % or 30%. What does this mean? is there any possible description of absolute values that can be compared with other apparatus? Under this aspect the work has to be improved .  It is not intuitive to understand the reason to move from a water dispersion of nanometric cylinders of CNC to a dry powder of circular almost micrometric  nanoparticles to then use the latter to do again a water dispersion with PVA. In my limited experience in nanocomposite materials I would have expected that the use of the pristine CNC dispersion would have performed better due to the rod like shape  of CNC (as reproted by the authors in Fig 4b).

All the discussion in line 408-417 is based on the SEM micrrographs  reported in Fihgure 19. To me. it an excessive speculation  for this simple figure.

Comments on the Quality of English Language

The work is not very easy to read, with many repeats 

Reviewer 2 Report

Comments and Suggestions for Authors

The presented paper offers a unique approach to CNC nanoparticle preparation via ultrasonic-assisted spray drying. The results and methodology are sound and reasonably explain the observed phenomena. The manuscript focuses on the preparation technique, so the authors provide many details, which undoubtedly will be attractive to the right audience.

I did not find any problems with the manuscript. Everything is carefully explained. To the best of my abilities, I did not find any issues with the quality of the English language.

Author Response

Thank you very much.

Reviewer 3 Report

Comments and Suggestions for Authors

In this manuscript, cellulose nanocrystals (CNCs) were prepared by the ultrasonic assisted spray dryer, and these nano spray dried CNCs (nano SDCNCs) were introduced into PVP to enhance its mechanical properties. It is indeed an exploration to use the ultrasonic assisted spray dryer to fabricate CNCs with desired particle size, yield, and production rate. The manuscript is well prepared with sufficient data and clear content organization supporting and presenting claims. Therefore, I would personally support its publication in Nanomaterials.

Author Response

Thank you very much.

Reviewer 4 Report

Comments and Suggestions for Authors

This manuscript relates to the application of ultrasonic assisted spray dryers to CNFs and CNCs to test whether ultrasonic assisted spray dryers can successfully dry these cellulosic materials. However, this paper is not suitable for publication in this form, which needs to be rewritten in light of the recommendations mentioned below.

Some major concerns are shown as following:

1. Please check the references carefully and then unify the format of the references.

2. Please check the full text carefully and change the following similar questions, for example: 1) There should be Spaces between numbers and units. 2) Please unify the unit format, some writing “min”, while some writing “minutes” in the manuscript. 3) “ml/min” should write as “mL/min”.

3. The formulas in the manuscript should be marked with serial number (1) (2) (3).

4. Scales should be added to Fig 4 (a), Fig 5, and Fig 16.

5. Introduction: It is necessary to clearly state in this part how this research differs from the research in the literature, that is, the authors should include the novelty of the manuscript.

6. In order to facilitate readers to understand the research content of this paper, the mechanism analysis and schematic diagram should be added.

7. The key indicators of the materials prepared in this paper should be compared with those in the literatures in tabular form.

8. The following two articles are well written and authors are advised to read and cite them.

1) Lu, J., Bai, T., Wang, D. et al. Electrospun Polyacrylonitrile Membrane In Situ Modified with Cellulose Nanocrystal Anchoring TiO2 for Oily Wastewater Recovery. Adv. Fiber Mater. (2023). https://doi.org/10.1007/s42765-023-00325-0

2) Wang, L., Fu, Q., Yu, J. et al. Cellulose Nanofibrous Membranes Modified with Phenyl Glycidyl Ether for Efficient Adsorption of Bovine Serum Albumin. Adv. Fiber Mater. 1, 188–196 (2019). https://doi.org/10.1007/s42765-019-00010-1

Comments on the Quality of English Language

The English language level of the article is good.

Round 2

Reviewer 1 Report

Comments and Suggestions for Authors

The authors have fulfilled the requests and now the manuscript can be accepted, in my opinion, for publication

Comments on the Quality of English Language

no special comment

Reviewer 4 Report

Comments and Suggestions for Authors

The authors have revised the manuscript carefully based on referees' comments. The scientific quality of this paper is greatly improved. The manuscript can be accepted at the present version. 

Comments on the Quality of English Language

The quality of english language is good